A novel steganography method for binary and color halftone images

Çiftci Efe 1 efeciftci@cankaya.edu.tr
Sümer Emre 2
1 Department of Computer Engineering, Çankaya University , Ankara , Turkey
2 Department of Computer Engineering, Başkent University , Ankara , Turkey
Akleylek Sedat
Electronic publication date: 2022 Aug 16
Publication date: 2022
Volume: 8
Electronic Location ID: e1062
Received 2022 Jan 28; Accepted 2022 Jul 18
Copyright: © 2022 Çiftci and Sümer
Copyright year: 2022
Copyright holder: Çiftci and Sümer
License: This is an open access article distributed under the terms of the Creative Commons Attribution License, which permits unrestricted use, distribution, reproduction and adaptation in any medium and for any purpose provided that it is properly attributed. For attribution, the original author(s), title, publication source (PeerJ Computer Science) and either DOI or URL of the article must be cited.
License URL: https://creativecommons.org/licenses/by/4.0/

Keywords: Steganography, Halftone image, Image processing, Plaintext payload, Secret sharing

Funding: The authors received no funding for this work.

==============================
Digital steganography is the science of establishing hidden communication on electronics; the aim is to transmit a secret message to a particular recipient using unsuspicious carriers such as digital images, documents, and audio files with the help of specific hiding methods. This article proposes a novel steganography method that can hide plaintext payloads on digital halftone images. The proposed method distributes the secret message over multiple output copies and scatters parts of the message randomly within each output copy for increased security. A payload extraction algorithm, where plain carrier is not required, is implemented and presented as well. Results gained from conducted objective and subjective tests prove that the proposed steganography method is secure and can hide large payloads.

Introduction

Steganography is the practice of hidden communication using unsuspicious media as carriers (Cheddad et al., 2010). The basis of this communication is that the presence of this communication should be known only by the participating parties while keeping everybody else unaware of this communication. In order to establish such hidden communication, steganographic methods require both a payload and a carrier cover media. Depending on the type of cover media, the payload is hidden onto the cover media by suitable algorithms. The result of this process is stego media, which is expected to look and feel exactly like a regular media and raise no suspicions.

While secret communication has been available since the oldest days of known human history, the advancements in digital communication have opened many new ways for steganography in the digital domain. In the digital domain, file types such as images, sounds, and videos are the most commonly used carriers for various steganography methods. Steganography methods for these carriers can be enhanced using additional techniques such as payload encryption or payload compression to obtain better results in terms of payload security, increased payload capacity, and improved stego media quality (Sharma & Batra, 2021; Sari et al., 2019; Sharma et al., 2019).

Digital image carriers can be mainly classified into (a) color images, (b) grayscale images, and (c) binary images. Color images are most commonly represented with 24 bits per pixel (red, green, and blue channels, 8-bits per channel), though 8-bit (indexed) and 32-bit (RGB + 8-bit alpha channel) color images are also available (Shreiner & Group, 2009). Pixels in grayscale images are represented with eight bits that store intensity information.

Binary images have only black and white tones; therefore, only one bit is enough to represent a pixel in binary images (0 – black, 1 – white). Binary images can be created from grayscale source images by either (a) thresholding (Al-Amri, Kalyankar & Khamitkar, 2010; Huang & Chau, 2008; Wang, Chung & Xiong, 2008) or (b) halftoning (Chan & Chen, 1998; Shiau & Fan, 1994; Knuth, 1987; Floyd, 1976). Halftone images are crafted from grayscale images through specific methods to achieve a look similar to continuous-tone grayscale images (Lau & Arce, 2001; Ulichney, 1987). They are most commonly preferred for printing in newspapers and books, where they help save ink by printing a black and white only image instead of more expensive grayscale images. Halftone images also occupy less space on storage devices, and they compress better as each pixel in the halftone image needs only one bit to be represented, as opposed to at least eight bits for grayscale or color images.

Inspired by color images with three channels, binary halftone images can also be improved to contain three separate binary channels to represent red, green, and blue color information in them. These types of images are called color halftone images and are produced by performing the regular halftoning procedure individually on all channels of a source color image. Because these images use three separate binary channels for each pixel, these images can represent colors with eight different values (Fig. 1). When compared to binary halftone images, the primary advantages presented by color halftone images are the increased capacity for payloads and lesser payload visibility due to the human visual system’s poor perception of similar colors (e.g., yellow and white).

Figure 1 Colors in a three-channel color halftone image.

Due to the facts, such as there is a wide range of previously published steganography methods for color and grayscale images (as will be discussed in the Related Work section), and also few discovered examples that hide plaintext payloads rather than other types of payloads (such as images, as proposed by John Blesswin et al., 2020; Yan et al., 2015, etc); the steganography method that is proposed in this article focuses on binary and color halftone images as carriers and hides plaintext payloads in them. The payload is hidden into the carrier during conversion of the carrier from grayscale/color into halftone format. The method is implemented to work on halftone images that are generated via either predefined patterns or error diffusion.

Although it would simplify the steps required for both embedding and extraction, hiding the payload in a single image may allow attackers to easily gain access to the whole payload with proper attack methods (Luo et al., 2021; Quach, 2014; Jiang et al., 2005). In order to improve the security of the payload for such cases, the proposed method takes inspiration from the secret sharing method (Naor & Shamir, 1995) and scatters portions of the payload into multiple slightly different output images. This way, it is ensured that the payload can be successfully extracted if and only if all output copies are collected back and processed together again. The results gained through conducted objective and subjective tests have shown that the proposed method can hide large plaintext payloads successfully without causing many disturbances on the carrier media. The length of the maximum payload is directly proportional to the spatial resolution of the cover media.

In addition to the hiding algorithm, a suitable payload extraction algorithm has also been implemented and presented in this article. In contrast to some existing methods, our extraction algorithm can successfully recover the payload without the need for the plain version of the carrier. This ensures a potential weakness is avoided where an attacker may attempt to extract the payload by comparing the plain and stego media.

The remainder of this article is organized as follows. In Related Work, we share some of the existing related and remarkable works in the steganography domain. In Materials & Methods, we explain both the proposed hiding algorithm (with its variations for mentioned carrier types and halftoning methods) and the payload extraction algorithm. In Experiments & Results, we present the conducted objective and subjective experimental tests and their results. In Discussions, we discuss the results of the experiments and highlight significant outcomes of these experiments. Finally, in the Conclusions, we summarize the presented study and state possible future studies related to the proposed method.

Related work

Digital images with higher bits per pixel (bpp) ratios, such as color and grayscale images, provide foundations for a wide variety of steganographic methods. The most common steganography method for color and grayscale images is the LSB method (Cheddad et al., 2010), where the payload is hidden into the least significant bit (or more than one bit in some cases) of each byte. The changes caused by this method are usually insignificant to the human visual system but can easily be detected by computers. Therefore, variations of the LSB method that aim to evade being detected by software by utilizing additional methods such as encryption have since been published (Kordov & Zhelezov, 2021; Zhou et al., 2016; Juneja & Sandhu, 2014, 2013; Hsiao, Chan & Chang, 2009; Sutaone & Khandare, 2008).

Steganography on color images is not limited to methods that operate on the least significant bit. For example, Nilizadeh et al. (2022) and Nilizadeh et al. (2017) propose methods that can hide any type of payload into the blue channel using matrix patterns generated from the green channel of an image. Mowafi et al. (2019) proposes a method that can hide plaintext payloads into an image’s Cb and Cr components using matrix patterns generated from the Y component. Color and grayscale images offer many possibilities, but the full review of color and grayscale images is out of this paper’s scope.

Since binary images do not offer as many features as grayscale or color images, methods that aim to hide in halftone images differ from the methods that use color or grayscale carriers; this has resulted in the proposal of new different methods that especially exploit the distinct structure of binary images. Cruz et al. (2018) proposes a method in which letters, digits, and punctuation marks are represented with unique 3 × 3 patterns, and the pattern form of a plaintext payload is distributed in cover media appropriately. Since the embedding process on halftone images causes more visible distortions than grayscale or color images, several methods such as those proposed by Yu et al. (2021), Lu et al. (2019), Xue et al. (2019) have been developed to minimize these distortions. Some methods, such as those proposed by Fu & Au (2003, 2001a), Pei & Guo (2003), require a plain (i.e., does not carry payload) halftone cover media during payload extraction. However, on the other hand, there are other methods, such as those proposed by Rosen & Javidi (2001), that do not require the plain cover media for payload extraction.

Several works published so far focus on improving the output quality of visual cryptography, which is a method for hiding visual payloads. Naor & Shamir (1995) propose the secret sharing method, where the pixels in the payload are scattered over multiple copies of the output (i.e., shares). Instead of keeping a secret message in one place, secret sharing aims to make it difficult for attackers to access the whole secret message directly and to ensure the security of the secret message by splitting it into subparts that will not make sense on their own and sharing them among more than one person. In the mentioned method, all shares must be stacked over to extract the payload. Wang, Arce & Di Crescenzo (2006) propose a method where the pixels of a binary payload image are hidden and distributed in an amount of generated halftone images using visual cryptography. Their method aims to improve the output quality by encoding the pixels using direct binary search method (Analoui & Allebach, 1992) to decrease the noise caused by visual cryptography. Fu & Au (2001b) propose two methods, named intensity selection and connection selection, that aim to improve the visual quality of carrier outputs generated by error diffusion algorithms by choosing the best locations for hiding the payload. Of these methods, intensity selection offers better visual quality, while connection selection has lower computational complexity than intensity selection. Some methods that focus on visual cryptography suffer from image expansion, in which a 1 × 1 white or black pixel in the payload gets to be represented by a larger block of pixels (e.g., 2 × 2) in the output images. Several methods, such as those proposed by Askari, Heys & Moloney (2013) and Chen et al. (2007), overcome this problem and enable both the cover and the payload images to be the same size.

Materials and Methods

The proposed algorithm (Algorithm 1) requires a source image I, a plaintext payload P, and the number of desired output images NSHARES as inputs. The algorithm adopts the previously explained secret sharing methodology; it produces a number of (i.e., NSHARES) output halftone images Iht and distributes each bit of the payload (i.e., B) in calculated positions over a randomly chosen output image (i.e., R) among all output images. This procedure produces multiple output images that have slight differences but still share an identical look. By distributing the payload this way, it is ensured that attackers will not be able to successfully extract the payload if they are missing even just a single output image.

Algorithm 1 Payload hiding.

 Input: source image (I), plaintext payload (P), number of shares (NSHARES)	
 Output: a set of stego images (Iht)	
1:  procedure PAYLOADHIDING(I, P, NSHARES)	
2:   dLen ← I.capacity/Pbinary.length	
3:   if dLen < 1 then   ▶ terminate if payload is larger than	
4:    return        ▶ the capacity	
5:   end if	
6:   let Iht be a set of NSHARES-long empty images	
7:   for each dLen-long blocks in I as block; do   ▶ choose a proper position for each	
8:    R ← randomly chosen output image   ▶ payload bit among all output images	
9:    RR ← randomly chosen (x, y) position in block   ▶ and hide it using one of the four	
10:    B ← Pbinary.next     ▶ provided methods depending on the	
11:    Iht (R,RR) ← embed(R, block, RR, B)    ▶ carrier type and halftoning method	
12:   end for	
13:  end procedure	

Aside from the secret sharing mechanism explained above, the hiding procedure also ensures that all bits in a given output image are spread across the image and are not concentrated in a specific region. This is achieved by calculating a distance length dLen; which is the ratio of the number of usable pixels in the cover image to the length of the payload. The image is then divided into groups of dLen sequential pixels (i.e., block), and each bit is embedded into a different group. Furthermore, the positions of these bits are randomized within each group (i.e., RR) in order to prevent detection from statistical attacks that target every Nth pixel. As a result of this randomization, the algorithm does not produce deterministic outputs; a different set of output images will be produced on each execution, although the inputs did not change.

The embed() function in Algorithm 1 denotes the four methods that have been implemented specifically for carrier types (i.e., gray or color) and halftoning method (i.e., halftoning via patterns or error diffusion). In order to hide a payload bit; the chosen output carrier and position (x, y) where the payload bit will be embedded must be determined earlier. Details of each embedding method are explained in the following sections.

Hiding on halftone images using patterns

This method requires a number of binary patterns for creating the halftone image. For this purpose, 10 3 × 3 binary patterns p0−9 have been defined as demonstrated by Zhang (2017) (Fig. 2).

Figure 2 (A–J) Binary patterns used for constructing halftone images.

Pixels in a regular 8-bit grayscale image contain 256 different intensity levels. In order to determine which pattern will be used for which intensity level in the source image I, these levels must be divided into 10 groups, and a simpler version of the source image (i.e., Igrouped) must be generated according to Eq. (1).

(1) Igrouped(i,j)=⌊I(i,j)26⌋

The left-hand side of Eq. (1) is a set of values in the range of 0 and 9, and the values of pixels in the mentioned simpler version consist of these values. Then, the final halftone image will be created by matching the value of each pixel with the matching pattern. The stages of this process are demonstrated in Fig. 3. It should be noted that since every pixel in the original image is now represented with a 3 × 3 pattern, the spatial resolution of output images is larger (e.g., 256 × 256 grayscale images are converted into 768 × 768 halftone images).

Figure 3 Stages of converting a grayscale image into pattern-based halftone image.

(A) Source image, (B) Simpler image, (C) Final image.

In order to hide the payload, we propose to alter this conversion process such that when a “0” or “1” bit is to be hidden in a pixel, the pattern prior or next to the actually determined pattern is used in a randomly chosen output image, while the determined pattern is still used in the same position in all other output images (Eq. (2)). Since neighboring patterns are involved in this process, our algorithm avoids hiding bits in pure black or white regions in the image as there is no pattern prior to p0 or next to p9, respectively. This precaution also prevents the generation of visible noise in smooth black or white regions of the produced stego media.

(2) Iht(R,i,j)=(p(Igrouped(i,j)−1),ifbit=0p(Igrouped(i,j)+1),ifbit=1p(Igrouped(i,j),ifIgrouped(i,j)=0orIgrouped(i,j)=9

Sample pairs of binary and color output images generated with the mentioned method are presented in Fig. 4. The length of the embedded payload is 2,048 bytes.

Figure 4 Pattern method results on (A–B) binary and (C–D) color images.

Hiding on halftone images using error diffusion method

Error diffusion is a popular halftoning method in which the residual error of each processed pixel is distributed to its neighboring pixels, creating a smoother appearance and a closer appearance to the original image in the process. In order to distribute the error, a coefficient filter that tells which neighboring pixel will receive how much of the error must be used. There have been numerous methods proposed with different filters; Floyd-Steinberg (Floyd, 1976), Shiau-Fan (Shiau & Fan, 1994), and Jarvis-Judice-Ninke (Jarvis, Judice & Ninke, 1976) are a few among the most popular. For example, the Floyd-Steinberg filter is demonstrated in Fig. 5.

Figure 5 Floyd-Steinberg coefficient filter.

We have previously explained that in order to hide a 0 or 1 bit, the proposed method chooses one random output share among others, and encodes a slightly different pattern than the ones used in the rest of the other share images. In order to adapt this method to error diffusion, we have simplified this process since we no longer have a set of 10 patterns to choose from. Instead, when a 0 or 1 is to be hidden, the value of the chosen pixel in one random output image is set to the desired value, while the pixels in the same coordinate in all the other output images are set to the opposite (Eq. (3)).

(3) Iht(R,i,j)={0,ifbit=01,ifbit=1

Sample pairs of binary and color output images generated with the mentioned method are presented in Fig. 6. The length of the embedded payload is 2,048 bytes.

Figure 6 Error diffusion method results on (A–B) binary and (C–D) color images.

Payload extraction

The hiding methods explained earlier scatter each payload bit to a randomly chosen output image in a fashion similar to the secret sharing methodology, so that the whole payload is rendered inaccessible without access to all output images. These bits are embedded in such a way that the pattern or color at the determined position for each bit is represented differently in the chosen output image than the rest of the images; so whenever a bit is hidden, an irregularity among the outputs appears. The extraction algorithms (Algorithms 2 and 3) operate by seeking these irregularities among all provided output images (SHARES). If all the patterns (for pattern-based carriers) or colors (for error diffusion carriers) at the same coordinates have been found to be the same, it is assumed that no bits were hidden at that position, and that position is skipped without performing any further operations. However, if an irregularity is detected, a single 0 or 1 bit is extracted from that position depending on the visual relationship of the outlier and regular media, and the extracted bit is appended to a bit string. When this operation is over, the obtained bit string is converted into ASCII characters to reveal the payload.

Algorithm 2 Payload extraction on pattern based carriers.

 Input: a set of stego media (SHARES)	
 Output: extracted plaintext payload	
1: procedure PAYLOADEXTRACTION(SHARES)	
2:  let bitString be empty string	
3:  for i = 1 → SHARESany.height do	
4:   for j = 1 → SHARESany.width do	
5:    for k = 1 → SHARESany.channels do	
6:     patterns ← ⊘	
7:     for l = 1 → SHARES.length do    ▶ collect the patterns at the same	
8:      patternsl,k ← SHARESl,i, j,k     ▶ (i, j) location from all images	
9:     end for	
10:     if count(unique(patterns)) = 2 then    ▶ extract a 1 or 0 bit	
11:      if count(prev(patterns)) > count(next(patterns)) then    ▶ if a different pattern	
12:       bitString += “1”    ▶ is detected in the	
13:      else     ▶ set of previously	
14:       bitString += “0”    ▶ collected patterns	
15:      end if	
16:     end if	
17:    end for	
18:   end for	
19:  end for	
20:  return ASCII(bitString)	
21: end procedure	

Algorithm 3 Payload extraction on error diffusion based carriers.

 Input: a set of stego media (SHARES)	
 Output: extracted plaintext payload	
1: procedure PAYLOADEXTRACTION(SHARES)	
2:   let bitString be empty string	
3:   for i = 1 → SHARESany.height do	
4:    for j = 1 → SHARESany.width do	
5:     for k = 1 → SHARESany.channels do	
6:      pixels ← ⊘	
7:      for l = 1 → SHARES.length do    ▶ collect the pixels at the same	
8:       pixelsl,k ← SHARESl,i, j,k    ▶ (i, j) location from all images	
9:      end for	
10:      if count(unique(pixels)) = 2 then     ▶ extract a 1 or 0 bit	
11:       if count(black(pixels)) > count(white(pixels)) then     ▶ if a different pixel	
12:        bitString += “1”    ▶ is detected in the	
13:       else     ▶ set of previously	
14:        bitString += “0”    ▶ collected pixels	
15:       end if	
16:      end if	
17:     end for	
18:    end for	
19:   end for	
20:   return ASCII(bitString)	
21:  end procedure	

As explained previously, all produced outputs for a given payload must be available for successful extraction. Otherwise, missing outputs cause a cascaded shift in extracted bits, resulting in illegible outputs (Fig. 7).

Figure 7 Outputs of payload extraction attempts with different numbers of inputs.

Results

In order to obtain results from the methods mentioned above, several tests that consist of embedding payloads of different lengths into different cover media have been conducted. The chosen cover media are airplane80, beach09, and forest22 images (Fig. 8) from UC Merced Land Use Dataset (Yang & Newsam, 2010).

Figure 8 Three test images from UC Merced Land Use Dataset.

(A) airplane80, (B) beach09, (C) forest22.

According to the previously explained payload hiding methods, the maximum payload capacity of these images is calculated at about 8,000 bytes for grayscale carriers and 24,000 bytes for color carriers. Multiple payloads have been generated using the Lipsum generator1 , which are large enough to fill 25% and 50% of the binary outputs of chosen images. In order to observe the effects of the payloads on different images, the same payloads are used for both binary and color halftone images. These tests have been repeated three times to produce separate sets of 4, 8, and 12 output images, respectively. The output images created during these tests are shared online2 for detailed inspection. In order to obtain a better quality assessment of the proposed method, the outputs obtained from these tests have been evaluated both objectively and subjectively.

Objective testing

For objective testing, SNR (Eq. (4)), PSNR (Eq. (5)) (Salomon, Motta & Bryant, 2007) and structural similarity (SSIM) (Eq. (6)) (Wang et al., 2004) values of each individual output for a selected test image and payload have been calculated. SNR and PSNR focus on the effects of added noise on the quality of the signal between two images (regular and stego media in our case), while SSIM focuses on perceptual differences between these images according to three key factors: luminance l, contrast c, and structure s. These metrics are used to compare modified digital images with their original counterparts to measure the differences between them and to evaluate the overall quality of the modified image.

(4) SNR=10log10(PsignalPnoise)

(5) PSNR=10log10(MAX2MSE)

(6) SSIM(x,y)=[l(x,y)]α⋅[c(x,y)]β⋅[s(x,y)]γ

Mean values are calculated from the values obtained from metrics mentioned above and presented in Figs. 9–12 and in Tables 1–4. The legends in these figures contain abbreviations of the carrier (i.e., “air” for airplane80, “bea” for beach09, “for” for forest22), the metric (i.e., snr, psnr, ssim), and the percentage of cover media the payloads are filling (i.e., 25p, 50p). The x and y axes represent the number of shares generated (i.e., 4, 8, 12) and calculated SNR, PSNR, and SSIM values, respectively.

Figure 9 (A–I) SNR, PSNR and SSIM values of pattern-generated binary halftone images.

Figure 10 (A–I) SNR, PSNR and SSIM values of pattern-generated color halftone images.

Figure 11 (A–I) SNR, PSNR and SSIM values of error diffusion-generated binary halftone images.

Figure 12 (A–I) SNR, PSNR and SSIM values of error diffusion-generated color halftone images.

Table 1 SNR, PSNR and SSIM values of pattern-generated binary halftone images.

	airplane80	beach09	forest22	
NSHARE	%	SNR	PSNR	SSIM	SNR	PSNR	SSIM	SNR	PSNR	SSIM	
4	25	19.40	21.62	0.98	18.68	21.58	0.98	17.67	21.58	0.98	
	50	16.39	18.61	0.96	15.67	18.57	0.96	14.66	18.57	0.96	
8	25	22.41	24.63	0.99	21.69	24.59	0.99	20.68	24.59	0.99	
	50	19.41	21.62	0.98	18.68	21.58	0.98	17.67	21.58	0.98	
12	25	24.17	26.39	0.99	23.46	26.35	0.99	22.44	26.35	0.99	
	50	21.17	23.39	0.98	20.44	23.34	0.98	19.43	23.34	0.98	

Table 2 SNR, PSNR and SSIM values of pattern-generated color halftone images.

	airplane80	beach09	forest22	
NSHARE	%	SNR	PSNR	SSIM	SNR	PSNR	SSIM	SNR	PSNR	SSIM	
4	25	24.04	26.40	0.99	23.34	26.35	0.99	22.23	26.35	0.99	
	50	21.04	23.40	0.98	20.33	23.34	0.98	19.22	23.34	0.98	
8	25	27.06	29.41	0.99	26.35	29.36	0.99	25.24	29.36	0.99	
	50	24.06	26.41	0.99	23.34	26.35	0.99	22.23	26.35	0.99	
12	25	28.82	31.17	0.99	28.11	31.13	0.99	27.00	31.12	0.99	
	50	25.82	28.17	0.99	25.10	28.12	0.99	23.99	28.11	0.99	

Table 3 SNR, PSNR and SSIM values of error diffusion-generated binary halftone images.

	airplane80	beach09	forest22	
NSHARE	%	SNR	PSNR	SSIM	SNR	PSNR	SSIM	SNR	PSNR	SSIM	
4	25	9.97	12.14	0.86	9.24	12.13	0.86	8.39	12.14	0.86	
	50	7.49	9.66	0.76	6.83	9.72	0.76	5.98	9.74	0.77	
8	25	12.98	15.15	0.93	12.25	15.14	0.93	11.40	15.15	0.93	
	50	10.50	12.67	0.88	9.84	12.73	0.88	9.00	12.75	0.88	
12	25	14.74	16.91	0.95	14.01	16.90	0.95	13.16	16.91	0.95	
	50	12.27	14.43	0.92	11.60	14.49	0.92	10.76	14.51	0.92	

Table 4 SNR, PSNR and SSIM values of error diffusion-generated color halftone images.

	airplane80	beach09	forest22	
NSHARE	%	SNR	PSNR	SSIM	SNR	PSNR	SSIM	SNR	PSNR	SSIM	
4	25	14.66	16.94	0.95	14.00	16.95	0.95	12.99	16.93	0.95	
	50	12.13	14.41	0.92	11.66	14.62	0.92	10.64	14.58	0.92	
8	25	17.66	19.94	0.97	17.02	19.97	0.97	16.00	19.95	0.97	
	50	15.13	17.41	0.96	14.69	17.65	0.96	13.65	17.60	0.96	
12	25	19.42	21.70	0.98	18.77	21.72	0.98	17.75	21.70	0.98	
	50	16.90	19.17	0.97	16.44	19.40	0.97	15.41	19.36	0.97	

Subjective testing

In order to gain subjective results alongside the results discussed in the previous section, a survey was conducted on a group of 95 people. In this survey, attendees were presented with 12 pairs of images that consisted of regular and stego versions of the test images and were asked to answer how much difference they could detect between both versions of all pairs at first glance. Table 5 shows the results gained from the mentioned survey.

Table 5 Results of subjective testing.

	Pattern-based binary	Error diffusion binary	
	airplane80	beach09	forest22	airplane80	beach09	forest22	
No differences	26%	25%	41%	12%	18%	26%	
Few differences	54%	61%	47%	45%	35%	49%	
Many differences	20%	14%	12%	43%	47%	25%	
	Pattern-based color	Error diffusion color	
	airplane80	beach09	forest22	airplane80	beach09	forest22	
No differences	49%	52%	56%	45%	40%	48%	
Few differences	42%	39%	39%	41%	45%	45%	
Many differences	8%	9%	5%	14%	15%	7%	

Comparison with existing methods and safety tests

Since the proposed method uses halftone images as carriers, none of the existing popular LSB or similar steganalysis methods (e.g., offered by tools such as StegExpose3 ) can produce reliable estimations about the payload even when the outputs are converted back to pseudo-grayscale via a gaussian filter. Due to the lack of steganalysis methods in the known literature that aim for halftone plaintext carriers, an alternative attack method has been implemented instead. In this method, the produced outputs are tested for resistance against extraction attempts with missing shares.

Since the algorithm hides the payload in bits, we have previously explained that a cascade extraction error is expected to occur even when a single output image is missing; unintelligible characters will be extracted in these cases instead. In order to prove this is true for all attempts, multiple extraction attempts have been performed on the output images created during objective tests. Each extraction attempt started with one share, and the number of provided shares was increased until all shares were present (e.g., 1 of 8, until 8 of 8). As a result of these tests, it has been observed that unless all shares were present during tests, the maximum length of coincidentally revealed and intelligible pieces of the payload is always less than 2% of the total length of the payload. As an example, extracted bytes of the same payload from the forest22 image with various shares have been presented in Fig. 13.

Figure 13 Extracted bytes from forest22 image using (A) three of four shares, (B) seven of eight shares, (C) 11 of 12 shares.

Discussion

Conduction of objective and subjective tests has made the evaluation of the quality of the method in different aspects possible. One common finding obtained from both evaluations is that the overall quality of color stego images is higher than their binary counterparts. On the other hand, objective and subjective evaluations produced several different results in different aspects.

From Figs. 9–12, it can be seen that the quality of produced stego output images increases when the length of payload decreases, the number of produced output images increases, or color images are used instead of binary images. Table 5 shows that the visibility and detection risk of the payload is lower in color images and images generated with patterns.

The values calculated from quality assessment methods in objective evaluation are completely consistent with each other; there are no outlier cases such as shorter payloads causing lower scores even once. Also, it has been found that heterogeneous images generally scored better than homogenous images. This finding is supported by subjective evaluation results as well: answers from the participants clearly show that the percentage of detected differences is always lesser in heterogeneous images. From this objective evaluation result, it can be deduced that cover images with large regions of heterogeneous textures (such as forest22) prove to be better cover images for our method.

As an interesting finding, most of the participants scored images generated with patterns higher than the ones generated with the error diffusion method. This finding is also present in objective results: metric scores of pattern-generated carriers are higher than their error diffusion-based counterparts. We believe the reason behind this is because alterations happening in patterns affect only one pixel in a 3 × 3 group of nine pixels, but they may affect every single pixel directly in images generated with error diffusion methods; the objective dissimilarity and subjective visibility of payload are higher in these images. From this result, it can be deduced that halftone images generated from patterns prove to be better cover images than images generated from error diffusion methods for our method.

When tested for payload extraction, it has been observed that when a large number of shares are present (e.g., 11 of 12), short sequences of letters from the payload may appear in random positions of the extracted text. However, they are never long enough to reveal meaningful information. Since the proposed methods are nondeterministic and produce new different sets of output images on each execution, the tests have been repeated multiple times to verify that the exposed information is never long enough to reveal a meaningful payload.

Conclusions

In this article, a novel steganography method that operates on halftone cover images is proposed and demonstrated. In general, halftone images offer a cheaper alternative to grayscale images in aspects of being resource effective both in printed and digital media. They are also immune to numerous LSB steganalysis methods that target grayscale and color images.

The method hides given plaintext payloads and distributes them on multiple outputs. The secret sharing approach has proven to be an effective aid for both decreasing the detectability risk of hidden payloads and also preventing attackers from successful payload extraction.

The method has been tested with different test images and with plaintext payloads of different lengths. The experimental results show that our method provides high embedding capacity for any given cover image. Furthermore, results obtained from both objective and subjective measures show that our method can produce outputs mostly indistinguishable from their unmodified counterparts and perform better on pattern-generated cover images. The results obtained from objective, subjective, and payload extraction tests indicate that the proposed method is suitable for real-life use.

This study mainly focuses on the presentation of the proposed method; the robustness of the method against steganalysis attacks is out of scope and has not been thoroughly tested. Also, a comparative analysis of the proposed algorithm with other algorithms could not be included as the implementation, inputs, or outputs of discovered methods are different from the proposed methods.

In order to further reduce detectability on halftone cover media with less spatial heterogeneous features, it is planned to implement a mechanism that proposes the maximum safest payload length according to spatial features of chosen cover media. Furthermore, as the proposed extraction algorithm currently operates only on digital carriers, it is also planned to improve the method for successful extraction from printed carriers as well.

Supplemental Information

Supplemental Information 1 MATLAB source codes for proposed method.

Click here for additional data file.

Additional Information and Declarations

Competing Interests

Author Contributions

Data Availability

1 Lorem Ipsum—All the facts—Lipsum generator; https://www.lipsum.com/

2 Halftone Steganography Results—Efe ÇİFTCİ; http://academic.cankaya.edu.tr/˜efeciftci/halftone-stego/

3 StegExpose; https://github.com/b3dk7/StegExpose

The authors declare that they have no competing interests.

Efe Çiftci conceived and designed the experiments, performed the experiments, analyzed the data, performed the computation work, prepared figures and/or tables, authored or reviewed drafts of the article, and approved the final draft.

Emre Sümer analyzed the data, authored or reviewed drafts of the article, and approved the final draft.

The following information was supplied regarding data availability:

The codes for the proposed message hiding and extraction methods were developed and tested in MATLAB 2021a and are available in the Supplemental Files.

Sample images in the Supplemental Files are chosen from UC Merced Land Use Dataset: http://weegee.vision.ucmerced.edu/datasets/landuse.html.

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
