# Peer review of "A novel steganography method for binary and color halftone images"

_PeerJ Computer Science, doi:10.7717/peerj-cs.1062_

## Round 0.1 · original submission · Major Revisions

The highlights and the novelty of the work are not clear. The methods used in the paper have already been addressed in the literature. Moreover, the subject is a very studied one in the literature, similar approaches have been previously published so many times in the conferences. Thereby, the paper is not suitable for publication in this format. More details are needed.

Reviewer 1 has suggested that you cite specific references. You are welcome to add it/them if you believe they are relevant. However, you are not required to include these citations, and if you do not include them, this will not influence my decision.

We encourage you to revise your manuscript and submit it again. Please address all comments in the revised version.

Reviewer 1 ·

Basic reporting

This work presents a new steganography method for halftone images. I discuss some of the structural, written English, and missing related works problems in the following.

Problem 1: English

Reading the paper is easy, and written English is good, but it has some grammar and typos which should be considered. The main problem is not using "the" in many cases, which was necessary. Please check the English with some tools. You can find some of the following mistakes below:

"Also, comperative analysis of proposed algorithm with other algorithms could not be included " should change to "Also, comparative analysis of the proposed algorithm with other algorithms could not be included"


" The method hides given plaintext payloads and distribute them on multiple outputs." should change to " The method hides given plaintext payloads and distributes them on multiple outputs."

"The length of maximum payload is directly" should change to "The length of the maximum payload is directly"

Spells of "diffusioning" and "comperative" are wrong.

"we discuss results of the experiments and highlight significant outcomes of these experiments. Finally, in Conclusions," should change to "In Discussions, we discuss the results of the experiments and highlight significant outcomes of these experiments. Finally, in the Conclusions"

Problem 2: Missing some close works

This work is basically based on the matrix pattern steganography method for RGB images that already some works are published. It is necessary to discuss them briefly in the paper, including:

"Adaptive Matrix Pattern Steganography on RGB Images"
"Steganography on RGB Images Based on a" Matrix Pattern" using Random Blocks"
"Information Hiding in RGB Images Using an Improved Matrix Pattern Approach."
"Combined algorithm of steganography with matrix pattern and pixel value difference"
"Image steganography using ycbcr color space and matrix pattern"
"A novel steganography method based on matrix pattern and LSB algorithms in RGB images"


This paper discussed that smooth neighborhoods could not generate matrix patterns that are previously discussed in the following papers. So, it is necessary to cite them including:

"Block-based reversible data embedding"
"Block Texture Pattern Detection Based on Smoothness and Complexity of Neighborhood Pixels"
"A Block Complexity based Data Embedding"

Problem 3: unclear graphs and notations

The other problem is with linear charts. It should be mentioned what are "x" and "y" in charts.

It is necessary to define all notations you used in codes. For example, what are "R," "RR," or "B"?

Experimental design

The idea of the paper is interesting, and it is discussed well in the paper, and it is great that this paper comes with a tool.

The main problem of this work is a minimal experimental study, as I understand only a small number of images are used, while in recent years, massive datasets have been used for steganography.

I have some questions:

1) Is any larger dataset available for these images' format?

2) Is the output of this tool deterministic? (The same picture with the same secret message always will have the same stego-image.)

3) Are matrix patterns selected manually, which are shown in Fig2? Or do you have any mechanism that can select matrix patterns randomly or select them based on the texture of the cover image?

4) Did you use ten patterns for hiding secret messages?

Validity of the findings

The idea is not novel, but it is enough for being accepted. I like how the paper is written, but it needs some improvement. The data are based on a small dataset, but it comes with a tool that shows the work is reliable.

However, it does not show that it can always work well with all images with different textures. This work can be extended by using a large dataset and studying which features and textures are suitable for this steganography method.

·

Basic reporting

no comment

Experimental design

no comment

Validity of the findings

no comment

Additional comments

According to the authors, the proposed steganography method in this work is secure and can hide large payloads.
1-However, when a new method is developed, its reliability and superiority/weakness over other methods must be strictly tested.
2-The authors said that they used the secret sharing method in steganography methods. However, the abstract not included this topic and was not adequately explained in the general flow of the article.

In order for this study to be published, it is obligatory to perform the above-mentioned examination, comparison, safety tests and additions.

---

## Round 0.2 · Major Revisions

One of the reviewers has pointed out important issues in the paper. Security analysis has several unclear points. We kindly ask you to revise the paper according to the reviewers' comments.

Reviewer 1 ·

Basic reporting

No comment

Experimental design

No comment

Validity of the findings

No comment

Additional comments

Thank you for updating the paper. It is clearer after answering questions and editing the paper. The English of the paper was good, and it is much better now. I like the algorithm, and I understand there is a lack of work for steganalysis in this work.

Also, when you are citing Mowafi et al. work in line 47, it is better to cite the following matrix pattern steganography beside it that is published in CVPR. Mowafi et al. used the CVPR paper and used Cb and Cr components.
Information Hiding in RGB Images Using an Improved Matrix Pattern Approach.

·

Basic reporting

no comment

Experimental design

no comment

Validity of the findings

no comment

Additional comments

This work is present a new steganography method for halftone images. But the novelty of the work is not clear.
The security criteria I suggested in Review 1 have not been adequately discussed.
I think it is not appropriate to publish this article in your journal.

·

Basic reporting

The basic presentation is appropriate. One of the improvement can be separation of related work from introduction section.

Experimental design

I was looking for the reason, why image is preferred by the author as cover file. Claims like, "Due to wide availability....." should an appropriate reference or justification for such claims. In statement, "We are inspired by secret sharing....", it is expected to specify the reason in term of quantifiable parameters. embed() function in algorithm 1 should be supported with a valid prototype and task. Comments can be used in algorithm for more clarification. An algorithm is expected to have a suitable name, input that it require and expected output.

Validity of the findings

It would be better to have separate graph for SNR and PSNR like as for SSIM. I was looking for few lines that explain the variation of these parameters according to the number of shares. While comparing with existing methods, comparison is expected with the recent work rather than the traditional LSB. Moreover, the comparison is expected on the same parameters used for evaluation. The SSIM attained is more than 0.95 in most of the cases that shows a great perceptual similarity between the images but I doubt that the same observation is not supported by the subjective testing.

Additional comments

The average values of PSNR ,SNR and SSIM achieved should be specified. Also discuss if the method developed is suitable for real use with those attained values of PSNR,SNR and SSIM.

·

Basic reporting

Recent literature references need to be cited.

Experimental design

No comments

Validity of the findings

No comments

---

## Round 0.3 · accepted · Accept

The reviewers are satisfied with the revision. Thus, we are happy to inform you that your manuscript has been accepted for publication with minor issues. Please check the language.

·

Basic reporting

The basic presentation is appropriate

Experimental design

no comment

Validity of the findings

no comment

Additional comments

The authors have revised the article in accordance with the referees's comments. In addition, they have supported the proposed method with various metrics (SSIM, SNR, PSNR, etc.). They have sufficiently discussed their method with subjective and objective tests. However, as the authors stated in the article, steganalysis was not performed in this study.
As a result, I think this article is appropriate to publish in your journal.

·

Basic reporting

No Comments

Experimental design

No Comments

Validity of the findings

No Comments

Additional comments

Authors may provide few lines as future scope. Applicability of this method on video steganography can be explored. Here are few of the suggested references which can be cited in this paper if author find it relevant.
Pilania, U., Tanwar, R., & Gupta, P. (2022). An ROI-based robust video steganography technique using SVD in wavelet domain. Open Computer Science, 12(1), 1-16.
Tanwar, R., Pilania, U., Zamani, M., & Manaf, A. A. (2021). An Analysis of 3D Steganography Techniques. Electronics, 10(19), 2357.

·

Basic reporting

Include Research Motivation in Introduction section
Include future scope in conclusion section

Experimental design

All previous comments have been addressed and manuscript seems to be fine

Validity of the findings

Authors have compared their proposed methodology with current state of art and seems fine